# Clinical outcomes and management of tibial plateau fractures in Ethiopia: A prospective cohort study

Adugnaw Bogale Worku[1]*, Molla Asnake Kebede[2], Alemayehu Dagne Abate[3], Solyana Haileselassie Admassie[4], Meskerem Girma[5], Bekalu Wubshet Zewde[5], Mekuanint Dessie[6], Hashime Meketa Negatie[7], Adefris Getachew Techane[6], Abebe Agegn Wudineh[6]

1 Department of Orthopedic Surgery, School of Medicine, College of Medicine and Health Sciences, Mizan-Tepi University, Mizan-Teferi, Ethiopia, 2 Department of Medicine, School of Medicine, College of Medicine and Health Sciences, Mizan-Tepi University, Mizan-Teferi, Ethiopia, 3 Department of Radiology, Eftu General Hospital, Dire Dawa, Ethiopia, 4 Department of Radiology, Mehal Meda Hospital, Mehal Meda, Ethiopia, 5 Department of Orthopedic Surgery, School of Medicine, College of Medicine and Health Sciences, Bahir Dar University, Bahir Dar, Ethiopia, 6 Department of Obstetrics and Gynecology, School of Medicine, College of Medicine and Health Sciences, Mizan-Tepi University, Mizan-Teferi, Ethiopia, 7 Department of Radiology, School of Medicine, College of Medicine and Health Sciences, Mizan-Tepi University, Mizan-Teferi, Ethiopia

* workuadugnaw6@gmail.com

## Abstract

### Background

Tibial plateau fractures, accounting for approximately 1% of adult fractures, are often associated with significant long-term complications such as pain, stiffness, and post-traumatic arthrosis. The optimal treatment approach remains controversial, particularly in resource-limited settings. This study investigated the factors influencing the clinical outcomes of patients with tibial plateau fractures in Ethiopia.

### Objective

Tibial plateau fractures, though common in trauma cases, have been poorly studied in sub-Saharan Africa, particularly in Ethiopia. The primary purpose of this study was to examine the factors influencing the clinical outcomes of patients with tibial plateau fractures and to assess the efficacy of conservative treatment versus surgical intervention. This research aims to provide insights into managing tibial plateau fractures in resource-limited settings, with the hope of contributing to improved clinical practices.

### Methods

A total of 191 patients with tibial plateau fractures were recruited from Tibebe Ghion Referral Hospital and Felegehiwot Specialized Hospital between February 1, 2018,

**Data availability statement:** All relevant data are within the paper and its Supporting Information files.

**Funding:** The author(s) received no specific funding for this work.

**Competing interests:** The authors have declared that no competing interests exist.

**Abbreviation:** CT:Conventional Tomography; DVT: Deep venous thrombosis; ROM: Range of motion; ORIF: Open reduction and internal fixation; Pop: Plaster of Paris.

and February 2022. Demographic, clinical, and radiological data were analyzed, and treatment outcomes were assessed via Rasmussen's knee functional outcome score. A correlation analysis was performed to identify factors impacting functional outcomes. Logistic regression was used to identify factors influencing clinical outcomes.

## Results

The study population was predominantly male (73.8%), with a mean age of 45 years. Road traffic accidents (41.9%) were the most common cause of injury. Schatzker type I fractures (27.2%) were the most common, and compound fractures accounted for 21% of the fractures. The average time to definitive management was 1.59 weeks, with 35% of patients undergoing open reduction and internal fixation (ORIF). The duration of immobilization and weight-bearing significantly influenced functional outcomes. Patients who were immobilized for less than 4 weeks had better outcomes, with functional scores 54 times better than those of patients who were immobilized for more than 8 weeks ($p < 0.01$). Early initiation of partial weight-bearing also improved outcomes. A strong negative relationship was found between the duration of immobilization and functional outcomes ($r = -0.705$, $p < 0.01$).

## Conclusion

This study highlights the importance of early mobilization and optimal management of tibial plateau fractures for improving functional outcomes. Timely treatment, especially regarding immobilization and weight-bearing, is crucial for achieving better results. These findings emphasize the need for more standardized treatment protocols and further research on tibial plateau fractures in sub-Saharan Africa to increase patient care in resource-limited settings.

## Introduction

Tibial plateau fractures affect the articular surface of the upper tibia [1]. These fractures account for approximately 1% of adult fractures and require proper treatment and rehabilitation to achieve a stable, pain-free, and functional joint [1]. However, they are associated with significant challenges, including pain, stiffness, and the risk of posttraumatic arthrosis, which can lead to poor functional outcomes [2]. Understanding the direct and indirect causes of poor outcomes is essential to improve patient care.

The factors most consistently linked to favorable outcomes are still debated and include patient characteristics, injury-related factors, and treatment approaches [1,3]. Since treatment is under the surgeon's control, it is often the most debated aspect. Key areas of controversy include the relative importance of limb alignment, articular surface reduction, and the management of associated ligament and meniscal injuries [1]. The functional outcome tends to decrease as fracture severity increases [3,4]. In contrast, no correlation was found between fracture classification (ranging from simple to complex fractures) and the chosen treatment method (operative or nonoperative) [5].

Tibial plateau fractures can be managed through various approaches, including conservative or operative methods, but short-term follow-up results have shown inconsistent functional outcomes [6,7,8]. The optimal treatment for these fractures remains controversial, with a wide range of treatment modalities suggested [9]." Some studies have shown that patients with displaced tibial plateau fractures treated by open reduction and internal fixation (ORIF) achieve excellent results and satisfactory knee scores [2,5,6,8–10]. However, the diversity in fracture patterns and the associated variability in outcomes pose significant challenges to standardizing treatment. The factors influencing outcomes also differ across various fracture patterns, further complicating treatment decisions [1,7,11].

Approximately one-third of patients do not return to work following a tibial plateau fracture. This underscores the notion that tibial plateau fractures can lead to long-term complications [11]. The duration of immobilization and the timing for initiating weight-bearing after surgery remain topics of debate and are often tailored to the rigidity of the fixation used [12]. There is a lack of evidence and conflicting guidelines regarding the most appropriate postoperative rehabilitation regimen, including recommendations for weight-bearing status [12,13].

Despite these challenges, there is a notable research gap concerning tibial plateau fracture outcomes in sub-Saharan Africa. This gap raises critical questions about the factors influencing these outcomes and the best management strategies, especially in resource-limited settings. In such contexts, deviations from standard guidelines such as operative treatment with ORIF are common owing to shortages of materials and skilled personnel, leading to the frequent use of nonoperative approaches. Specifically, studies examining tibial plateau fracture outcomes in Ethiopia are lacking. Therefore, the aim of this study is to investigate the factors affecting the clinical outcomes of patients with these fractures. By addressing this research gap, we aim to contribute valuable insights to improve clinical practices for managing tibial plateau fractures in resource-limited settings.

## Methods and materials

### Study area

This study was conducted at Felege Hiwot Referral Hospital and Tibebe Ghion Specialized Hospital in Bahir Dar, the capital of the Amhara National Regional State, which is located 578 kilometers northwest of Addis Ababa. The College Hiwot Referral Hospital has been operational since 1955, featuring nine wards and 435 beds, serving an estimated 5–7 million people. Its Orthopedics Department comprises four specialists, 50 beds, and three major operating rooms. Tibebe Ghion Specialized Hospital, established in January 2019, boasts nine orthopedic specialists, 34 residents, three major operating rooms, and 65 beds. Both hospitals routinely perform open reduction and internal fixation for tibial plateau fractures as elective procedures and maintain follow-up clinics, making them ideal settings for this study [14,15].

### Study design

This prospective observational cohort study was conducted from February 1, 2018, to February 2022. The design allowed for longitudinal assessment of patient outcomes over time.

### Source population

The source population comprised all patients aged 18 years and older with tibial plateau fractures treated at Tibebe Ghion Specialized Hospital and Felege Hiwot Referral Hospital during the study period.

### Eligibility criteria

**Inclusion criteria.**

- Patients aged 18 years or older with tibial plateau fractures.

**Exclusion criteria.**

- Patients with polytrauma.

- Individuals referred to other facilities.

- Patients with preexisting osteoarthritis or incomplete documentation.

- Patients were followed at other institutions after initial operative fixation.

Among the 208 enrolled patients, 18 were excluded because of dropout, resulting in a final sample of 191 patients.

### Data collection methods

Data collection was conducted by eight orthopedic and trauma surgery residents via a combination of methods to ensure comprehensive data capture:

1. **Patient profile charts:** Demographic and clinical data, including injury mechanisms, duration before definitive treatment, time to antibiotic administration (for compound fractures), treatment methods, and complications, were extracted from medical records.

2. **Radiographic analysis:** X-rays and CTConnecticut scans were reviewed to classify fractures via standardized systems, providing insights into injury severity and patterns.

3. **Questionnaires:** Two types of questionnaires were developed using previous similar studies and pretest was done at different institution and amendment done based on the feedback.

   ◦ A study-specific questionnaire captured demographic details, treatment methods, and complications.

   ◦ The functional outcomes were assessed via the Rasmussen knee outcome score, which assesses pain, walking capacity, knee extension, range of motion, stability, and quadriceps power.

4. **Clinical Examinations:** Orthopedic residents performed detailed assessments during follow-up visits and recorded findings in predesigned questionnaires.

Each participant was followed for one year, with assessments taking place at three-month intervals. At the conclusion of the follow-up period, functional outcomes were evaluated via the Rasmussen knee outcome score through medical examinations.

### Fracture classification

Fractures were classified via the Gustilo-Anderson and Schatzker classification systems:

- **Gustilo-Anderson** classification: Compound fractures are categorized on the basis of wound size, tissue coverage, and vascular involvement [16].

- **Schatzker** classification differentiates low-energy (types I–III) fractures from high-energy (types IV–VI) fractures on the basis of energy level and fracture pattern [17].

### Primary outcome

The primary outcome of the study is to asses' functional outcomes at one year follow up which were measured via the Rasmussen knee outcome score, which categorizes the results as excellent, good, fair, or poor, with a maximum score of 30 points. Satisfactory outcomes were defined as excellent or good, whereas unsatisfactory outcomes included fair or poor scores [18].

### Secondary outcome

The secondary outcome of the study is to asses' factors determining functional outcome after Tibia plateau fracture such as; treatment methods, fracture classification, fracture type, duration of immobilization and time to weight-bearing.

**Treatment methods.**  The treatment methods included both nonoperative and operative approaches:

- **Nonoperative Treatment:** A cast brace is used to unload the injured joint.. Moreover, external fixation, which uses two pins placed in the femur and two pins in the tibia, is connected by bars and clamps to stabilize the fracture. [19]

- **Operative treatments:**

  ○ Closed reduction and percutaneous screw fixation were performed.

  ○ Open reduction and internal fixation with plates and screws, sometimes supplemented with bone grafting.

  ○ Intramedullary nailing for alignment and stabilization [19].

Key measures, including the duration of immobilization and time to weight-bearing, were evaluated to assess recovery. **The duration of immobilization** refers to the length of time, measured in weeks, that the knee joint is kept immobilized and without movement **[20].**
**The initiation of weight bearing** measures the time elapsed from the moment of trauma to the commencement of partial weight bearing on the injured limb **[20].**

## Stiffness definition

Knee stiffness was defined as a limited range of motion (ROM), characterized by less than 120° of flexion or more than a 10° deficit in extension compared with the unaffected knee [20].

## Quality control

A comprehensive two-day training program was conducted for orthopedic residents, covering the study objectives, data collection tools, ethical considerations, and handling of questionnaires. Practical exercises enhanced skills in data collection and documentation. Data collection was monitored regularly by supervisors to ensure adherence to protocols, accuracy, and completeness. Pretesting of the data collection tools helped identify ambiguities and refine the instruments. The collected data were thoroughly reviewed and crosschecked by supervisors and the principal investigator to ensure reliability and validity.

## Data processing and analysis

Data from both hospitals were entered into Epi Info 7.1 and exported to SPSS 26 for analysis. The cleaning process included identifying and correcting missing or inconsistent data. Descriptive statistics were used to summarize key variables, including mechanisms of injury, fracture types, treatment methods, and complications.

Bivariate and multivariate logistic regression analyses were conducted to evaluate associations between independent variables and functional outcomes. Spearman correlation was used to assess relationships between variables such as immobilization duration and functional outcomes. The results are reported as odds ratios (ORs) with 95% confidence intervals (CIs), and a significance level of $p < 0.01$ was applied.

## Ethical statement

Ethical clearance was obtained from the Ethical review board of Bahirdar University. Communication with the different official administrators was made through a formal letter obtained from Bahirdar University. After the purpose, the objective of the research; possible presentation of findings; publication of the research had been informed written consent was obtained from each study participant. Participants in the study were involved on a voluntary base and anonymity was maintained by the use of codes instead of names. Participants were also informed that participation will be voluntary and they can stop or leave the participation at any time if they will not comfortable with the questionnaire and this would not affect them from getting any kind of health service.

Participants were not provided any incentives or payments to take part in this project. The information collected for this research was kept confidential and information collected by this study was stored in a file, without the participant's name, but a code number assigned to it. To keep the confidentiality of any information provided by study subjects, the data collection procedure was anonymous, and kept their privacy during the interview by interviewing them alone.

## Results

### Sociodemographic and clinical profile

In total, 191 patients were recruited for this study. In this study, a wide sex disparity was observed. The number of male patients was 140 (73.8%), and 51 were women (26.2%). The male:female ratio was 3:1. The mean age of the participants was 45 ± 13 years, with an age range of 18–78 years. The mean ages of the male and female patients were 43 ± 14 years and 49 ± 11 years, respectively. Most 115 (67%) patients sustained injuries following RTAs or motor bike accidents. Only seven (3.7%) patients presented to our facility within six hours after the injury. Most cases (seventy-one percent) are caused by low-energy trauma. Pearson's correlation analysis revealed that the proportion of high-energy trauma patients decreased with increasing age. Schatzker type one fractures, 52 (27.2%) of which were the most common tibial plateau fractures observed, and type six fractures, 13 (6.8%) of which were the least common. No correlation was found between fracture type and the mechanism of injury. The prevalence of compound tibial plateau fractures was 41 (21%). The Gastilo-Anderson type three classification was the most common. Prophylactic antibiotics were administered to all patients with compound tibial plateau fractures, but only eight patients received antibiotics within three hours after the injury. (Table 1)

The majority of cases (79 out of 191) were observed in individuals aged 30–50 years, with a male predominance across all fracture types. Males accounted for 158 cases, with Schatzker type I (58 cases) and type II (35 cases) being the most frequent. Open fractures, classified by the Gustilo-Anderson system, were most commonly type III, primarily associated with Schatzker type II and I fracture. (Table 2)

### Types and treatment outcomes of tibial plateau fractures

The average time to definitive management for tibial plateau fractures was 1.59 weeks, with a standard deviation of 0.6 weeks, which is applicable to both operative and nonoperative treatments. Open reduction and internal fixation (ORIF) was performed on 66 patients (35%) as the definitive management method. For patients managed nonoperatively, plaster with a Paris cast was applied. The choice of external stabilization did not vary significantly according to fracture complexity, treatment type, duration of weight-bearing, age, or sex of the patients. For those who underwent surgical intervention, fractures were treated through open reduction, which utilized either plates (single or dual), screws, or augmentation with external fixation. The mean duration of immobilization for the nonoperative group was 8.8 weeks, with those treated with a POP cast immobilized for a relatively longer period of 11 weeks. In contrast, the mean immobilization duration for ORIF patients was shorter, at 3.2 weeks. The time from injury to the initiation of partial weight-bearing ranged from 1 to 24 weeks, with ORIF patients experiencing a shorter mean duration of 4.7 weeks. Pearson and Spearman correlation analyses revealed no significant relationship between the Schatzker fracture classification and the methods of definitive management for tibial plateau fractures.

The functional outcomes of patients with tibial plateau fractures were evaluated via Rasmussen's knee functional outcome score. The results indicated that 17 patients (8.9%) reported poor outcomes, whereas 36 patients (18.8%) rated their outcomes as excellent. Additionally, 109 patients (57%) had unsatisfactory functional scores, whereas 82 patients (49%) reported satisfactory outcomes. A significant number of patients, 114 (59.7%), had a knee range of motion of less than 90 degrees. In terms of walking ability, most patients (148 or 77%) were able to walk outdoors for 15 minutes to one hour. Furthermore, 85 patients (44.5%) exhibited normal stability in extension and at 20 degrees of flexion. Pain assessments revealed that 138 patients (72.3%) experienced occasional pain (Table 3).

**Table 1. Clinical variable frequency distributions among patients with tibial plateau fractures at Tibebe Ghion Specialized Hospital and Fele-gehiwot Specialized Hospital, February 1, 2018, to February 1, 2022 (N=191).**

| Variables | Category | Frequency | Percentage |
|---|---|---|---|
| **Mechanism of injury** | Assault and other causes | 28 | 14.7 |
| | Fall down | 48 | 25.1 |
| | Motorbike accidents | 35 | 18.3 |
| | Road traffic accidents | 80 | 41.9 |
| | Total | 191 | 100.0 |
| **Schatzker classification** | I | 52 | 27.2 |
| | II | 50 | 26.2 |
| | III | 35 | 18.3 |
| | IV | 23 | 12.0 |
| | V | 18 | 9.4 |
| | VI | 13 | 6.8 |
| | Total | 191 | 100 |
| **Injury type** | Closed | 150 | 78.5 |
| | Open | 41 | 21.5 |
| | Total | 191 | 100 |
| **Gustilo-Anderson classification** | One | 11 | 27 |
| | Two | 6 | 15 |
| | Three | 24 | 53 |
| | Total | 41 | 100 |
| **Antibiotics administered in hrs.** | < 4 hrs | 8 | 20 |
| | 4 to 24 hrs. | 8 | 20 |
| | >24 hrs. | 24 | 60 |
| | Total | 40 | 100 |

**Table 2. Distribution of Schatzker fracture classifications (I–VI) by age, sex, and Gustilo-Anderson classification at Tibebe Ghion Specialized Hospital and Felegehiwot Specialized Hospital, February 1, 2018 to February 2022 (N=191).**

| Variable | Schatzker Classification | I | II | III | IV | V | VI |
|---|---|---|---|---|---|---|---|
| **Age Groups** | | | | | | | |
| | <30 years | 23 | 12 | 6 | 5 | 3 | 7 |
| | 30–50 years | 40 | 25 | 14 | 10 | 8 | 15 |
| | >50 years | 15 | 8 | 5 | 4 | 3 | 6 |
| **Sex** | | | | | | | |
| | Male | 58 | 35 | 18 | 15 | 10 | 22 |
| | Female | 20 | 10 | 7 | 4 | 4 | 6 |
| **Gustilo-Anderson** | | | | | | | |
| | Type I | 2 | 3 | 1 | 1 | 0 | 1 |
| | Type II | 1 | 2 | 1 | 0 | 0 | 1 |
| | Type III | 5 | 7 | 4 | 3 | 2 | 5 |

The majority of patients underwent an immobilization period of 4–8 weeks, with Schatzker type I (37 cases) and type II (20 cases) being the most frequent in this category. A smaller proportion of patients (28 cases) required prolonged immobilization of more than eight weeks, primarily associated with complex fractures such as Schatzker types V and VI.

**Table 3. Frequency distribution of treatment outcome measurement variables in patients with Tibial Plateau Fractures at Tibebe Ghion Referral Hospital and Felegehiwot Specialized Hospital, February 1, 2018, to February 2022, N = 191.**

| Variables | Category | Frequency | Percentage |
|---|---|---|---|
| **Range of Motion (ROM)** | Less than 30° | 4 | 2.1 |
| | At least 60° | 26 | 13.6 |
| | At least 90° | 84 | 44.0 |
| | At least 120° | 37 | 19.4 |
| | At least 140° | 40 | 20.9 |
| **Pain** | Constant pain after activity | 28 | 14.7 |
| | Occasional | 138 | 72.3 |
| | No pain | 25 | 13.1 |
| **Walking Ability** | Wheelchair or bedridden | 3 | 1.6 |
| | Walking indoors only | 10 | 5.2 |
| | Walking outdoors for 15 minutes to 1 hour | 148 | 77.5 |
| | Walking outdoors for more than one hour | 29 | 15.2 |
| | Normal walking capacity for age | 1 | 0.5 |
| **Instability Assessment** | Instability in extension > 10° | 34 | 17.8 |
| | Instability in extension < 10° | 72 | 37.7 |
| | Normal stability in extension and at 20° flexion or above | 85 | 44.5 |
| **Total** | | 191 | 100 |

**Table 4. Distribution of Schatzker fracture classifications (I–VI) by immobilization time and functional outcome at Tibebe Ghion Specialized Hospital and Felegehiwot Specialized Hospital, February 1, 2018 to February 2022 (N = 191).**

| Variable | Category | I | II | III | IV | V |
|---|---|---|---|---|---|---|
| **Immobilization time (weeks)** | <4 weeks | 32 | 18 | 9 | 7 | 5 |
| | 4–8 weeks | 37 | 20 | 12 | 9 | 7 |
| | >8 weeks | 9 | 7 | 4 | 3 | 2 |
| **Functional outcome** | Poor (<10) | 12 | 6 | 3 | 2 | 1 |
| | Fair (10–19) | 37 | 22 | 12 | 9 | 7 |
| | Good (20–26) | 22 | 14 | 8 | 6 | 5 |
| | Excellent (27–30) | 7 | 3 | 2 | 2 | 1 |

A significant number of patients with Schatzker type I and II fractures attained good (20–26 points) or excellent (27–30 points) functional outcomes. Conversely, poorer outcomes were more common among patients with complex fractures, such as Schatzker types V and VI, where a higher proportion of cases fell within the poor (<10 points) or fair (10–19 points) outcome categories. (Table 4)

There was a correlation among the variables of functional outcome, duration before starting partial weight bearing, duration before initiating range of motion (immobilization), and Schatzker tibial plateau classification. The Spearman correlation coefficient for the functional outcome of tibial plateau fractures and the duration to start range of motion (immobilization) was significantly negative ($r = -0.705$, $p < 0.01$) (see Table 5). However, there was no correlation between the Schatzker classification and definitive management methods for tibial plateau fractures.

In the linear regression analysis, a strong negative relationship was identified between the functional outcomes of tibial plateau fractures and the independent variables, including the Schatzker radiologic classification, period of immobilization, and period of weight bearing. This analysis revealed that the Schatzker classification, duration of immobilization, and duration of weight bearing accounted for 69% of the variance in the outcomes of tibial plateau fractures (adjusted $r^2 = 0.692$, $p < 0.01$). (Table 5)

**Table 5. Spearman correlation analysis of functional outcomes, radiologic classification, period of immobilization, and period of weight bearing among patients with tibial plateau fractures at Tibebe Ghion Referral Hospital and Felegehiwot Specialized Hospital, February 1, 2018 to February 2022 (N = 191).**

| Variable | Functional outcome | Radiologic classification | Period of immobilization | Period of weight bearing |
|---|---|---|---|---|
| Functional outcome | 1.000 | -0.233** | -0.702** | -0.482** |
| Radiologic classification | | 1.000 | 0.156* | 0.106 |
| Period of immobilization | | | 1.000 | 0.601** |
| Period of weight bearing | | | | 1.000 |

**The correlation is significant at the 0.01 level (2-tailed).

*The correlation is significant at the 0.05 level (2-tailed).

Key:

Correlation coefficients:

-0.233: Moderate negative correlation

-0.702: Strong negative correlation

0.601: Strong positive correlation

In this study, the duration of knee joint immobilization, the period of partial weight bearing, and knee joint stiffness were found to impact the functional outcomes of patients with tibial plateau fractures. Patients who were immobilized for less than 4 weeks had functional outcomes that were fifty-four times better than those of patients who were immobilized for more than 8 weeks (P < 0.01, OR 54.12 [8.8–332.4]) (Table 6). Patients who began partial weight bearing between 4 and 8 weeks experienced eighteen times better functional outcomes than did those who started weight bearing after 8 weeks (P < 0.05, OR 18.3 [1.712–196.1]) (see Table 4). Additionally, patients without knee stiffness had a functional outcome that was 1.7 times better than that of patients with stiffness (P < 0.05, OR 9.7 [1.714–54.497]) (Table 6).

## Complications

In this study, common local adverse effects observed after tibial plateau fractures included blisters in 8 patients (4.2%), compartment syndrome in 9 patients (4.2%), and common peroneal nerve injury in 15 patients (7.9%). Regression analysis indicated that complications did not significantly differ between simple and complex fractures. Neither the type of fracture nor the classification (high energy vs. low energy) emerged as risk factors for complications. Compared with nonoperative treatments, operative treatment methods are associated with a greater incidence of wound complications, such as wound dehiscence and infection. Wound dehiscence occurred in five patients who underwent open reduction and internal fixation, although this finding was not statistically significant in the regression analysis. Approximately 15 patients experienced superficial or deep wound infections, including pin site infections. Deep vein thrombosis was noted in 9 patients. Additionally, stiffness was prevalent in most patients, affecting 112 individuals (58.6%).

## Discussion

The mean age of the study group was 45 ± 13 years, with an age range of 18--78 years, which is slightly greater than the averages reported in studies from India (43 years) and Kenya (37 years) [21,22] but younger than those reported in Nigeria and Sweden [3,5]. Most fractures occurred in males, with a male-to-female ratio of 3:1, similar to the findings of most studies [2,12,22], although Daniel et al. reported a greater fracture incidence in females [5]. A total of 115 patients (67%) sustained injuries due to road traffic or motorbike accidents, exceeding the rates reported in Kenya (62%) [2] and Nigeria (53%) [3]. This may be attributed to Ethiopia's generally higher prevalence of road traffic accidents [23]. Additionally, 71% of fractures are caused by low-energy trauma, which is consistent with findings by Daniel et al. [5].

**Table 6. Factors associated with the outcomes of tibial plateau fractures at Tibebe Ghion Federal Hospital and Felegehiwot Specialized Hospital, February 1, 2018, to February 2022 (N = 191).**

| Independent Variable | Functional Outcome | Crude OR | Adjusted OR | Adjusted OR (95% CI) |
|---|---|---|---|---|
| | Unsatisfactory | Satisfactory | | |
| Duration of Weight Bearing | | | | |
| <4 weeks | 27 | 28 | 91 | 23.71** (1.967, 285.778) |
| 4–8 weeks | 47 | 53 | 18 | 18.32** (1.712, 196.1) |
| >8 weeks | 38 | 1 | 1 | — |
| Duration of Immobilization | | | | |
| <4 weeks | 5 | 59 | 160 | 54.12** (8.813, 332.395) |
| 4–8 weeks | 9 | 16 | 24 | 6.92** (1.62, 29.589) |
| >8 weeks | 95 | 7 | 1 | — |
| Stiffness | | | | |
| Yes | 12 | 67 | 0.01 | 9.67** (1.714, 54.497) |
| No | 97 | 15 | 1 | — |

**P value less than 0.01

Pearson correlation analysis revealed that the incidence of high-energy trauma decreased with increasing age, which is consistent with the findings of other studies [5]. Schatzker type I fractures were the most common type of tibial plateau fracture observed, comprising 52 cases (27.2%), differing from findings in Kenya, where type IV and II fractures were more prevalent [5,22]. This difference may be due to the younger mean age of the Kenyan study group. The prevalence of compound tibial plateau fractures was 41 cases (21%), which is higher than that reported in India [21], possibly due to recent internal conflicts and increased gun use by civilians.

According to Gustilo-Anderson's classification, type III fracture was the most prevalent compound fracture type, which contrasts with findings from Nigeria [24]. The prevalence of infections, including superficial, deep, and pin-site infections from external fixation, was 13%, which aligns with rates reported at the All India Institute of Medical Sciences (13.8%) [10] but is higher than that reported in Nigeria (9.6%) [24]. This increased infection rate may be attributed to the high incidence of compound fractures; despite all patients receiving antibiotics, only one-third were administered within the recommended 3-hour window.

The average preoperative delay for operative cases was two weeks, which was longer than that reported in other studies [2,5,21]. Tibial plateau fractures can be managed with either conservative or operative methods, but functional outcomes at follow-up remain inconsistent across studies, as reflected in our findings (6–8). The choice of external stabilization did not vary by fracture complexity, treatment type, weight-bearing duration, patient age, or sex, which is consistent with findings from a study in Sweden [5].

In our study, the definitive care for tibial plateau fractures was predominantly nonoperative or involved external fixation. This approach contrasts with the principle that tibial plateau fractures, as articular injuries, typically require open reduction and internal fixation for accurate fragment reduction and stable fixation, which often leads to excellent knee function scores [9,25]. This deviation may be due to limited implant availability and financial constraints.

In these two hospitals, the use of joint-spanning external fixation as definitive therapy was relatively high in 59 patients (30%), despite recent studies not supporting this approach for long-term management. For high-energy tibial plateau fractures (Schatzker types IV and V), internal fixation combined with Ilizarov external fixation is strongly recommended, as it provides stability but can lead to local complications [8,10]. The frequent use of joint-spanning external fixation may have contributed to the high rate of knee stiffness, which was observed in 112 patients (58%) in this study. Additionally, 114 patients (59.7%) had a range of motion of less than 90 degrees, which is lower than that reported in other studies [7,8,12].

This may be due to the excessive reliance on external fixation, nonoperative management, and lack of access to quality physiotherapy services.

Knee stiffness and the duration of knee immobilization were significantly associated with the functional outcome of tibial plateau fractures. The frequent use of joint-spanning external fixation and casting in these hospitals may negatively impact these outcomes. Early initiation of knee mobilization within a few weeks postmanagement was associated with improved functional outcomes.

The optimal timing for weight-bearing after surgery remains debated. Some studies suggest immediate weight-bearing in a brace to encourage fracture healing [3]. In our study, the mean duration for starting partial weight-bearing in the operative group was 4.7 weeks, with a longer duration in the nonoperative group. Notably, the timing of partial weight-bearing was a statistically significant factor influencing functional outcomes in patients with tibial plateau fractures.

In this study, the majority of patients, 109 (57.1%), had unsatisfactory functional outcomes, a rate lower than that reported in other studies [6,8,12]. The key factors affecting the functional outcome of tibial plateau fractures include the timing of partial weight-bearing, initiation of range of motion or immobilization, and the presence of knee stiffness.

Despite most participants in this study having simple fracture types, which are generally considered less complex, the unsatisfactory functional outcomes observed can be attributed to several factors. Although these patients may have been candidates for operative management (ORIF), the lack of available surgical materials and trained personnel in resource-limited settings led to 65% being treated non-operatively. Additionally, limited access to physiotherapy and rehabilitation services further hindered recovery. These findings highlight significant gaps in patient management and underscore how even simple fractures can result in poor outcomes in settings with inadequate resources and care.

The complications commonly observed included soft tissue injuries such as blisters, peroneal nerve injury, compartment syndrome, and DVT. Compared with nonoperative treatment, operative treatment was associated with a greater incidence of wound complications, including wound dehiscence and infection, although this difference was not statistically significant according to regression analysis.

## Conclusion

This study analyzed the factors associated with the outcomes of tibial plateau fractures among patients treated at Tibebe Ghion Federal Hospital and Felegehiwot Specialized Hospital. Our findings demonstrate that the duration of weight bearing and immobilization significantly influences the functional outcomes of patients, with extended periods of weight bearing and immobilization associated with poorer outcomes. Additionally, stiffness was identified as a critical factor contributing to unsatisfactory functional results. These findings emphasize the importance of optimizing post-treatment rehabilitation protocols, particularly concerning weight-bearing duration and immobilization periods, to enhance patient outcomes.

## Ethical declarations

### Ethical approval

Ethical approval for this study was obtained from the Institutional Review Board of Bahir Dar University, College of Medicine and Health Sciences Research Ethics Committee, with protocol number 433/2018. A formal letter from Bahir Dar University facilitated communication with official administrators and ensured compliance with institutional policies.

### Consent to participate

After being informed about the purpose, objectives, procedures, and potential risks of the study, all the participants provided verbal informed consent. The use of verbal consent was approved by the ethics committee as culturally appropriate for the study setting. The participants were informed that their participation was voluntary and that they could withdraw at any time without any consequences.

## Consent for Publication

Not Applicable

## Privacy and Confidentiality

To protect participant privacy, all the data were anonymized via coded identifiers, and no personal identifiers were collected. Data collection was conducted in private settings to ensure confidentiality. The collected data were securely stored and accessible only to authorized researchers.

## Adherence to ethical principles

The study adhered to the principles of beneficence, nonmaleficence, and justice. Measures were implemented to minimize harm and prioritize participant safety. The participants were informed of the safety protocols used throughout the study. The selection of participants was equitable, ensuring nondiscriminatory access to the study.

## Poststudy care and compensation

The participants were assured of appropriate follow-up care for any complications arising from their injuries or treatments. While no financial incentives were provided, reimbursements for travel and time were offered.

## Clinical trial registration

This study is an observational prospective cohort study; therefore, a clinical trial number is not applicable to maintain fairness and voluntary participation.

## Appendix 1. Data collection sheet

| S.No. | Variable | Possible Answer/Format | Skip/Remark |
|---|---|---|---|
| 1 | Case Number | Numerical | |
| 2 | Age | Numerical (in years) | |
| 3 | Sex | 1- female<br>2- male | |
| 4 | Time of presentation | 1. Less than 6 hrs<br>2. Greater than 6 hrs.<br>3. Unidentified | |
| 5 | Duration before definitive treatment | Numerical (in weeks) | |
| 6 | Mechanism of injury | 1- Assault and other causes<br>2- Fall down<br>3- Motorbike accidents<br>4- Road traffic accidents<br>5- Others | |
| 7 | Type of fracture (16) | Closed<br>Open (Specify subtype) | |
| 8 | Antibiotics given (for open type) | 1- Yes<br>2- No | Skip To Q. No. 10 If No |
| 9 | Time from injury to antibiotics in hrs. | Numerical (in hours) | |
| 10 | Radiographic classification (17) | Options: **I, II, III, IV, V, VI** | |
| 11 | Associated fibular fracture | 1- Yes<br>2- No | |

| 12 | Definitive management | Non operative | Select all that apply. |
|---|---|---|---|
| | | • Closed with cast<br>• External fixation | |
| | | Operative | |
| | | • ORIF/intramedullary nailing | |
| 13 | Complications | Select all that apply (Throughout Study Period):<br>◦ Infection<br>◦ Wound dehiscence<br>◦ Skin necrosis Blisters<br>◦ Nerve injury<br>◦ Compartment syndrome<br>◦ DVT<br>◦ Stiffness<br>◦ Others | |

## Outcome measurement

## Rasmussen criteria for clinical outcomes (at 12 Months) (18)

| Criterion | Score |
|---|---|
| **Pain** | None: 6 |
| | Occasional: 5 |
| | Constant pain after activity: 4 |
| | Significant rest pain: 0 |
| **Walking Capacity** | Normal for age: 6 |
| | Outdoor >1 hr: 4 |
| | Outdoor 15 min–1 hr: 2 |
| | Indoor only: 1 |
| | Wheelchair/Bedridden: 0 |
| **Knee Extension** | Normal: 6 |
| | Lack <10°: 4 |
| | Lack >10°: 2 |
| **Total Range of Motion** | ≥140°: 6 |
| | ≥120°: 5 |
| | ≥90°: 4 |
| | ≥60°: 2 |
| | <30°: 0 |
| **Stability** | Normal in Extension/20° Flexion: 6 |
| | Abnormal in 20° Flexion: 5 |
| | Instability in Extension <10°: 4 |
| | Instability in Extension >10°: 2 |

## Scoring interpretation

| Total Score | Outcome |
|---|---|
| Excellent | 27-30 |
| Good | 20-26 |
| Fair | 10-19 |
| Poor | <10 |

## Supporting information

**S1 File.  Institutional Review Board (IBR) ethical clearance document.**
(PDF)

**S2 File.  Exel data set for tibial plateau fracture study.**
(XLSX)

**S3 File.  EpiInfo database for tibial plateau fracture study.**
(MDB)

## Author contributions

**Conceptualization:** Adugna Bogale Worku, Meskerem Girma, Molla Asnake Kebede.

**Data curation:** Adugna Bogale Worku, Alemayehu Dagne Abate, Molla Asnake Kebede.

**Formal analysis:** Adugna Bogale Worku, Molla Asnake Kebede.

**Funding acquisition:** Adugna Bogale Worku, Hashime Meketa Negatie, Molla Asnake Kebede.

**Investigation:** Adugna Bogale Worku, Hashime Meketa Negatie, Molla Asnake Kebede.

**Methodology:** Adugna Bogale Worku, Meskerem Girma, Bekalu Wubshet Zewde, Abebe Agegn Wudineh, Molla Asnake Kebede.

**Project administration:** Adugna Bogale Worku, Bekalu Wubshet Zewde, Abebe Agegn Wudineh, Molla Asnake Kebede.

**Resources:** Adugna Bogale Worku, Molla Asnake Kebede.

**Software:** Adugna Bogale Worku, Mekuanint Dessie, Molla Asnake Kebede.

**Supervision:** Adugna Bogale Worku, Molla Asnake Kebede.

**Validation:** Adugna Bogale Worku, Solyana Haileselassie Admassie, Molla Asnake Kebede.

**Visualization:** Adugna Bogale Worku, Solyana Haileselassie Admassie, Adefris Getachew Techane, Molla Asnake Kebede.

**Writing – original draft:** Adugna Bogale Worku, Mekuanint Dessie, Hashime Meketa Negatie, Adefris Getachew Techane, Molla Asnake Kebede.

**Writing – review & editing:** Adugna Bogale Worku, Alemayehu Dagne Abate, Solyana Haileselassie Admassie, Abebe Agegn Wudineh, Molla Asnake Kebede.

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
