## [Decision Letter · Decision Letter 0]

6 Mar 2025

PONE-D-24-58406Clinical Outcomes and Management of Tibial Plateau Fractures in Ethiopia: A Prospective Cohort StudyPLOS ONE

Dear Dr. Worku,

Thank you for submitting your manuscript to PLOS ONE. After careful consideration, we feel that it has merit but does not fully meet PLOS ONE’s publication criteria as it currently stands. Therefore, we invite you to submit a revised version of the manuscript that addresses the points raised during the review process. Your letter is addressed to PLOS ONE. However, you intende to submit the manuscript to Springer International Orthopedics. Ist that a mistake ? Two experts in the field made the review. Please,addressse their comments and suggestions during the revision.

We look forward to receiving your revised manuscript.

Kind regards,

Hans-Peter Simmen, M.D., Professor of Surgery

Academic Editor

PLOS ONE

Journal Requirements:

3. In the online submission form, you indicated that [The data underlying the results presented in this study are available from the corresponding author upon reasonable request.].

Reviewers' comments:

Reviewer's Responses to Questions

**Comments to the Author**

1. Is the manuscript technically sound, and do the data support the conclusions?

Reviewer #1: Partly

Reviewer #2: Yes

2. Has the statistical analysis been performed appropriately and rigorously? 

Reviewer #1: Yes

Reviewer #2: No

3. Have the authors made all data underlying the findings in their manuscript fully available?

Reviewer #1: Yes

Reviewer #2: No

4. Is the manuscript presented in an intelligible fashion and written in standard English?

Reviewer #1: Yes

Reviewer #2: Yes

5. Review Comments to the Author

Reviewer #1: Although an interesting Manuscript, there are limitations and questions which need to be answered by the authors:

Q1. Predominantly Schatzker I fractures were included (27.2%?), Schatzker VI were only 6.8%.

Since Schatzker I is not a difficult fracture to treat, what is the reason that the funcional outcome is so poor?

Unsatisfactory outcome was reported with 57%, limited flexion of the knee joint below 90 Degrees with 57.9%

Q2. Section complications: compartment syndrome is reported with 4.2% and peroneal nerve injury with 7.9%.

These figures are quite high, even in contract with predominantly Schatzker I fractures. Are the listed complications

primary caused by the accident or related to surgery (postoperative complications)?

Recomendations:

1. Please put in a table the numbers and percentages of the Schatzer I, II, III, IV, V and VI Fractures, related to age, sex

and combidities (like Nicotine/ drug abuse, arterial and venous vascular disease, diabetes, renal disease).

Advisable is also to list the immobilisation time related tot the Schatzker classification

2. Please list in a table the Gusilo-Anderson classification related to the Schatzker Classification. This will provide more

information about the severity of the injury of the whole patient cohort

3. Please list in a table the functional outcome related to the Schatzker classification.

Reviewer #2: The authors describe clinical outcomes one year after tibial plateau fractues in a single center hostpital in Ethiopia. There are only few studies about this topic in Africa and therefore the topic is very important and novel to report.

In the reviewed manuscript line numbering and the tables are missing. This makes sound reviewing difficult. Nevertheless there are a few comments already implementable by the authors to improve the quality of the study at hand.

The newest reference is from 2019. There is much more literature about these fractures available, for example:

https://pubmed.ncbi.nlm.nih.gov/?term=Tibial+Plateau+Fractures+in+africa

The statistical testing is not done properly. Data need to be tested for normal distribution and thereafter are tested with the tests accordingly. Furthermore, why was mulivariate logistic regression performed? The power of the data sample is probably not allowing this test. Univariate analysis is of course possible. Simple t-test, chi square, fishers exact test and Mann-Whitney U test respectively should be accurate for the data sample.

Please adhere to the Strobe Checklist for reporting of your study

https://www.strobe-statement.org/checklists/

I further recommend defining a primary outcome: For example functional follow up at one year and define all other values as secondary outcomes. Therewith the very nicely conducted study could further gain value.

Further feedback is possible after line numbering and access to the tables. Thank you for your work.

6. PLOS authors have the option to publish the peer review history of their article (what does this mean? ). If published, this will include your full peer review and any attached files.

**Do you want your identity to be public for this peer review?** For information about this choice, including consent withdrawal, please see our Privacy Policy .

Reviewer #1: No

Reviewer #2: **Yes: ** Samuel Haupt

---

## [Author Response · Author response to Decision Letter 1]

9 Apr 2025

Reviewer 1

Although an interesting Manuscript, there are limitations and questions which need to be answered by the authors:

Q1. Predominantly Schatzker I fractures were included (27.2%?), Schatzker VI were only 6.8%.

Since Schatzker I is not a difficult fracture to treat, what is the reason that the funcional outcome is so poor? Unsatisfactory outcome was reported with 57%, limited flexion of the knee joint below 90 Degrees with 57.9%

Answer:

The poor functional outcomes, despite the majority of participants having Schatzker I fractures (27%), which are typically considered less complex to treat, can be attributed to several factors. Although patients with Schatzker I fractures may have indications for operative management, such as ORIF (Open Reduction and Internal Fixation), the lack of available surgical materials and trained personnel in resource-limited settings likely prevented many from receiving the appropriate treatment. As a result, 65% of patients were treated with non-operative management methods. Additionally, the limited access to adequate physiotherapy and rehabilitation centers further impacted the overall treatment outcomes.

The main goal of this paper is to highlight these gaps in care, which point to the inefficiency of patient management in resource-limited environments. These findings suggest that even simple fractures, such as Schatzker I, can result in poor functional outcomes when critical resources and skilled care are lacking.

We added on the discussion part

Q2. Section complications: compartment syndrome is reported with 4.2% and peroneal nerve injury with 7.9%.

These figures are quite high, even in contract with predominantly Schatzker I fractures. Are the listed complications primary caused by the accident or related to surgery (postoperative complications)? We didn’t identify whether these complication (compartment syndrome and peroneal nerve) were observed dueto surgery or due to, fracture.

Answer.

The reported complications of compartment syndrome (4.2%) and peroneal nerve injury (7.9%) appear relatively high, particularly given the predominantly Schatzker I fractures in our study. We fully acknowledge that it would have been beneficial to explicitly determine whether these complications were primarily due to the fracture itself or associated with the surgical intervention. Unfortunately, this distinction was not made in our study. As a result, it remains unclear whether the complications arose from the trauma of the fracture or as postoperative issues related to surgery.

Recomendations:

1. Please put in a table the numbers and percentages of the Schatzer I, II, III, IV, V and VI Fractures, related to age, sex and combidities (like Nicotine/ drug abuse, arterial and venous vascular disease, diabetes, renal disease).

Advisable is also to list the immobilisation time related tot the Schatzker classification

Answer; accepted and added

2. Please list in a table the Gusilo-Anderson classification related to the Schatzker Classification. This will provide more information about the severity of the injury of the whole patient cohort

Answer; accepted and added

3. Please list in a table the functional outcome related to the Schatzker classification.

Answer; accepted and added

Reviewer 2:

The authors describe clinical outcomes one year after tibial plateau fractues in a single center hostpital in Ethiopia. There are only few studies about this topic in Africa and therefore the topic is very important and novel to report.

Thank you!

1-In the reviewed manuscript line numbering and the tables are missing. This makes sound reviewing difficult. Nevertheless there are a few comments already implementable by the authors to improve the quality of the study at hand.

Answer, Accept and corrected.

2- The newest reference is from 2019. There is much more literature about these fractures available, for example:

https://pubmed.ncbi.nlm.nih.gov/?term=Tibial+Plateau+Fractures+in+africa

3- The statistical testing is not done properly. Data need to be tested for normal distribution and thereafter are tested with the tests accordingly. Furthermore, why was mulivariate logistic regression performed? The power of the data sample is probably not allowing this test. Univariate analysis is of course possible. Simple t-test, chi square, fisher’s exact test and Mann-Whitney U test respectively should be accurate for the data sample.

Answer: Our study included a total population of 191 participants, which we considered to be sufficiently large for the analyses performed. We initially conducted bivariate analysis for each independent variable and the outcome variable (functional outcome). Based on the statistical significance observed in the bivariate analysis, we proceeded with multivariate analysis to assess the associations between these variables. Additionally, Spearman's correlation was used to evaluate relationships between variables, such as immobilization duration and functional outcomes. We chose to perform multivariate logistic regression due to the significance observed in the bivariate analysis, and aimed to explore these associations more comprehensively. While univariate analyses, including t-tests, chi-square, Fisher's exact test, and Mann-Whitney U test, would also be appropriate for this dataset, we believed that the sample size was adequate to support the analyses conducted.

3- Please adhere to the Strobe Checklist for reporting of your study

https://www.strobe-statement.org/checklists/

Answer; accepted and adhered

4- I further recommend defining a primary outcome: For example functional follow up at one year and define all other values as secondary outcomes. Therewith the very nicely conducted study could further gain value.

Answer: Accept and Corrected. In methodology part.

---

## [Editor Report · Decision Letter 1]

17 Apr 2025

Clinical Outcomes and Management of Tibial Plateau Fractures in Ethiopia: A Prospective Cohort Study

PONE-D-24-58406R1

Dear Dr. Worku,

We’re pleased to inform you that your manuscript has been judged scientifically suitable for publication and will be formally accepted for publication once it meets all outstanding technical requirements. After revision your manuscript looks much better. The recommendations of both reviewers have been taken into account.

Kind regards,

Hans-Peter Simmen, M.D., Professor of Surgery

Academic Editor

PLOS ONE
---

## [Editor Report · Acceptance letter]

PONE-D-24-58406R1

PLOS ONE

Dear Dr. Worku,

I'm pleased to inform you that your manuscript has been deemed suitable for publication in PLOS ONE. Congratulations! Your manuscript is now being handed over to our production team.

Kind regards,

on behalf of

Dr. Hans-Peter Simmen

Academic Editor

PLOS ONE